# SweetDreamer: Aligning Geometric Priors in 2D Diffusion for Consistent Text-to-3D

**Weiyu Li**[1,2], **Rui Chen**[3,4,*] **Xuelin Chen**[4,†] **Ping Tan**[1,2,†]

[1] Hong Kong University of Science and Technology
[2] Light Illusions
[3] South China University of Technology
[4] Tencent AI Lab
{weiyuli.cn, xuelin.chen.3d}@gmail.com, riorui@foxmail.com, pingtan@ust.hk

## Abstract

It is inherently ambiguous to lift 2D results from pre-trained diffusion models to a 3D world for text-to-3D generation. 2D diffusion models solely learn view-agnostic priors and thus lack 3D knowledge during the lifting, leading to the multi-view inconsistency problem. We find that this problem primarily stems from geometric inconsistency, and avoiding misplaced geometric structures substantially mitigates the problem in the final outputs. Therefore, we improve the consistency by aligning the 2D geometric priors in diffusion models with well-defined 3D shapes during the lifting, addressing the vast majority of the problem. This is achieved by fine-tuning the 2D diffusion model to be viewpoint-aware and to produce view-specific coordinate maps of canonically oriented 3D objects. In our process, only coarse 3D information is used for aligning. This "coarse" alignment not only resolves the multi-view inconsistency in geometries but also retains the ability in 2D diffusion models to generate detailed and diversified high-quality objects unseen in the 3D datasets. Furthermore, our aligned geometric priors (AGP) are generic and can be seamlessly integrated into various state-of-the-art pipelines, obtaining high generalizability in terms of unseen shapes and visual appearance while greatly alleviating the multi-view inconsistency problem.

## 1 Introduction

Generative models have achieved diverse and high-quality image generation, in a highly controllable way with input text prompts (Nichol et al., 2022; Ramesh et al., 2021; Saharia et al., 2022b; Rombach et al., 2022). This remarkable achievement has been attained by training scalable generative models, particularly diffusion models, on an extensive corpus of paired text-image data. To replicate such success in 3D, a substantial endeavor is obviously necessary to gather a vast amount of high-quality text-3D pairs, which is currently receiving commendable attention (Deitke et al., 2023; Wu et al., 2023; Shrestha et al., 2022). However, it is evident that the effort required to collect a comprehensive 3D dataset covering highly varied subjects is considerably more significant, given the high cost associated with acquiring high-quality 3D content.

On the other end, attempts to achieve text-controlled 3D generative models have taken several routes, among which the 2D-lifting technique has emerged as a particularly promising direction and is increasingly gaining momentum in the field (Poole et al., 2022). This technique lifts 2D results into a 3D world and features an optimization framework, wherein a 3D representation is updated in differentiable parameterizations with the Score Distillation Sampling (SDS) loss derived from a pre-trained 2D diffusion model. By combining SDS with various suitable 3D representations (Lin et al., 2023; Chen et al., 2023; Wang et al., 2023b; Shi et al., 2023), this technique can generate high-fidelity 3D objects and scenes for a diverse set of user-provided text prompts.

Yet lifting 2D observations into 3D is inherently ambiguous. 2D diffusion models solely learn 2D priors from individual images and therefore lack 3D knowledge for disambiguation during the

---

*Work done during internship at Tencent AI Lab.
†Corresponding author

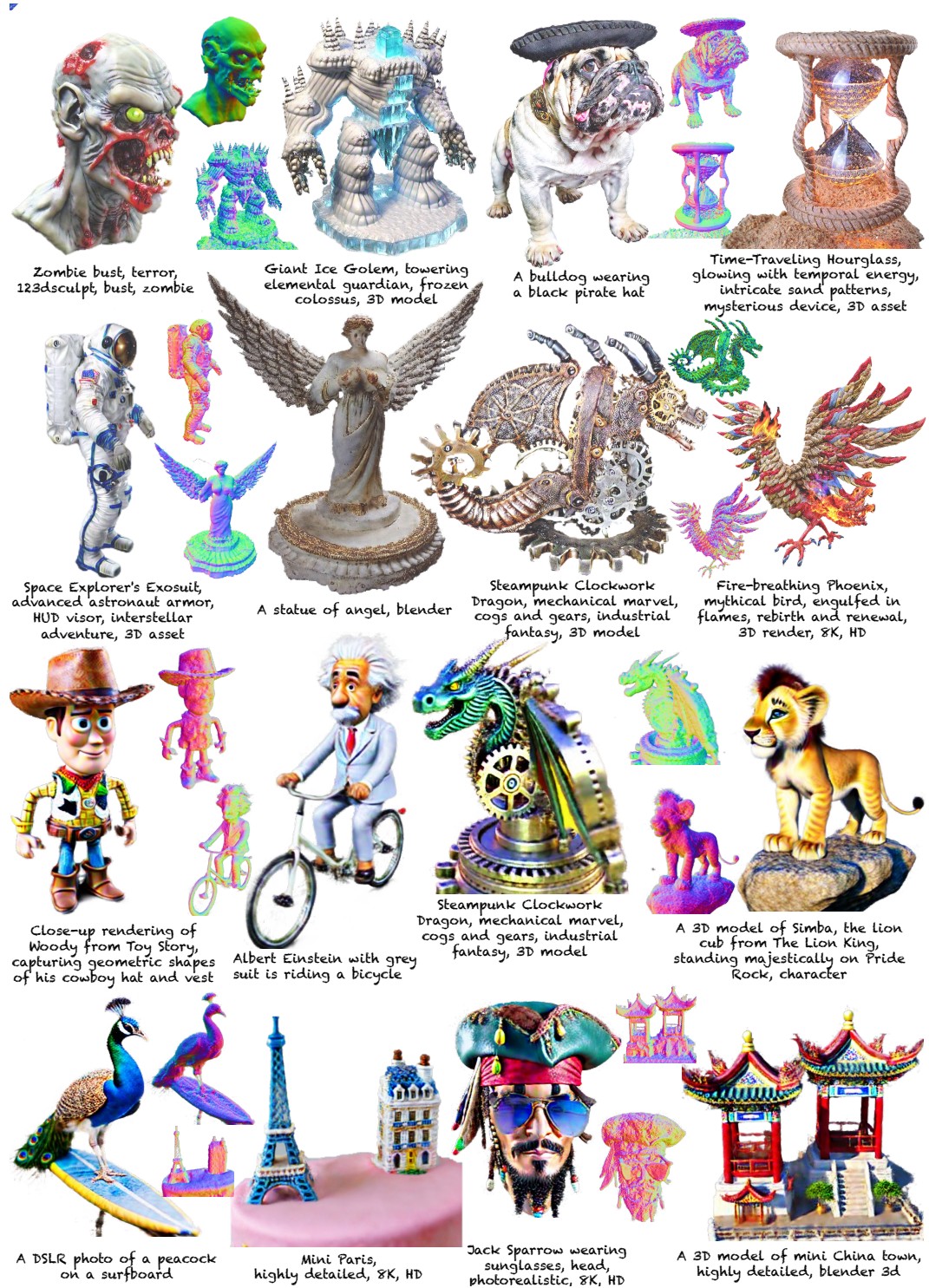

Figure 1: Our work can generate high-fidelity and highly diversified 3D results from various text prompts, free from the notorious multi-view inconsistency problem. We highly recommend referring to the supplementary materials for a more immersive viewing experience of the 3D results.

lifting, leading to notorious multi-view inconsistency problems, e.g., the multi-face Janus problem. While learning robust 3D priors from extensive and comprehensive 3D datasets is seemingly the very answer, in reality, we are only presented with 3D data that is rather scarce compared to plentifully available images. Hence, a currently compelling direction is to incorporate 3D priors learned from relatively limited 3D data into 2D diffusion priors that possess high generalizability, thus obtaining *the best of both worlds*.

In particular, the issues related to multi-view inconsistency can be primarily categorized into two types: i) geometric inconsistency issues, that are caused by the ambiguity in the spatial arrangement of geometric structures, i.e., a geometric structure can position and orient differently in 3D. Importantly, geometry inconsistency is further exacerbated during the lifting by the supervision imposed by 2D priors that *lack 3D awareness*, where many irrational 3D structures resulting in identical 2D projections can deceive 2D priors; ii) appearance inconsistency issues, that arise due to the ambiguity in the mapping from geometric structures to corresponding appearance, and again, is exacerbated by the lack of 3D awareness in 2D diffusion for disambiguation. Empirically, we found the geometry inconsistency issue is the primary cause contributing to most multi-view inconsistent results within various existing methods, whereas the appearance inconsistency issue manifests itself alone in only extreme cases and thereby holds lesser significance. This is evidenced by the fact that the majority of 3D inconsistent results exhibit repetitive geometric structures, typically multiple hands, or faces, that are generated under the guidance of 2D diffusion. It is worth noting that addressing these misplaced structures plays a significant role in mitigating 3D inconsistency in the final outcomes, as the inclusion of geometric hints from 3D consistent geometries greatly aids the appearance modeling. This holds true for both one-stage text-to-3D pipelines (Poole et al., 2022), where geometry and appearance are updated simultaneously, as well as pipelines that model geometry and appearance separately (Chen et al., 2023; Richardson et al., 2023). However, it should also be acknowledged that there may still be exceptional circumstances where appearance inconsistency can manifest with 3D consistent geometric structures.

These findings have motivated us to prioritize addressing geometric inconsistency issues in text-to-3D, by equipping the 2D priors with the capability to produce 3D consistent geometric structures while retaining their generalizability. In a way analogous to (Leike & Sutskever, 2023), we enforce the 2D geometric priors in diffusion models act in a way that *aligns with well-defined 3D geometries* as depicted in 3D datasets during the lifting, addressing the vast majority of the inconsistency problem from the origin. We refer to the resulting geometric priors as "AGP", for Aligned Geometric Priors. Specifically, we align the geometric priors by fine-tuning the 2D diffusion model to produce coordinate maps of objects in canonical space, thereby *disambiguating the geometry distribution* in 3D for ease of learning, and further conditioning it on additional camera specifications, thereby conferring *3D awareness* eventually. Notably, in stark contrast to methods that hinge heavily upon the geometric and appearance information in 3D datasets, we only capitalize on *coarse geometries*, avoiding over-reliance on geometric and visual appearance details that may further introduce undesired inductive bias. This "coarse" alignment of geometric priors not only enables the generation of 3D objects without the multi-view inconsistency problem but also retains the ability in 2D diffusion models to generate vivid and diversified objects unseen in 3D datasets.

Finally, yet importantly, our AGP possesses high compatibility that is generally lacking in competing methods. We show that AGP is highly generic and can be seamlessly integrated into various state-of-the-art pipelines using different 3D representations, obtaining high generalizability in terms of unseen geometries and appearances while significantly alleviating multi-view inconsistency. It represents a new state-of-the-art performance with a $85+\%$ consistency rate in human evaluation (see qualitative results gallery in Figure 1 and quantitative results in Table 1).

## 2 RELATED WORK

In the following, we mainly review related literature that exploit 2D priors learned in text-conditioned generative image models for text-to-3D, and refer readers to Zhang et al. (2023) for a more in-depth survey of text-to-image diffusion models.

**Text-to-3D using 2D Diffusion.** Following successful text-to-image diffusion models, there has been a surge of studies that lift 2D observations in diffusion models to perform text-to-3D synthesis,

bypassing the need for large-scale text-3D datasets for training scalable 3D diffusion models. In particular, the pioneer work by Poole et al. (2022) introduces a key technique – Score Distillation Sampling (SDS), where diffusion priors are used as score functions to supervise the optimization of a 3D representation. Concurrent with Poole et al. (2022), a similar technique is proposed in Wang et al. (2023a), which applies the chain rule on the learned gradients of a diffusion model and back-propagate the score of a diffusion model through the Jacobian of a differentiable renderer to optimize a 3D world. An explosion of text-to-3D techniques occurred in the community since then that improves the text-to-3D in various aspects, such as improved sampling shedules (Huang et al., 2023), adopting various 3D representations (Lin et al., 2023; Tsalicoglou et al., 2023b; Chen et al., 2023), new score distillation (Wang et al., 2023b), etc. Although these methods have shown the capability to circumvent the necessity for text-3D data and generate photo-realistic 3D samples of arbitrary subjects with user-provided textual prompts, more often than not, they are prone to the notorious 3D inconsistency issues. Hence, prior works have attempted to address the inconsistency with improved score functions and/or text prompts (Armandpour et al., 2023; Hong et al., 2023). Nonetheless, these methods cannot guarantee 3D consistency and tend to fail on consistent text-to-3D synthesis.

Concurrently, MVDream (Shi et al., 2023) addresses the multi-view inconsistency problem via training a dedicated multi-view diffusion model, which simultaneously generates multi-view images that are consistent across a set of sparse views. Particularly, they jointly fine-tune the diffusion model on real images and synthetic multi-view images, in order to inherit the generalizability in 2D diffusion and to obtain the multi-view consistency in 3D datasets. Instead of relying on computationally intensive renderings and fine-tuning on both synthetic and real images, our method uses only low-resolution and low-cost geometry maps, and hence the "coarse" alignment of geometric priors is computationally efficient. Besides, our AGP is generic and can be seamlessly integrated into existing pipelines to confer 3D consistency. Such compatibility is obviously lacking in MVDream.

**Generative Novel Views with Diffusion Models**   Another route of 3D generation is to model it as a view-conditioned image-to-image translation task with diffusion models, and directly generate novel views of the 3D, detouring the need for optimizing a 3D world. Watson et al. (2022) train a pose-conditional image-to-image diffusion model to take a source view and its pose as inputs and generate a novel view for a target pose. This method has only been demonstrated on synthetic data in the ShapeNet dataset (Chang et al., 2015). Similarly, Zhou & Tulsiani (2023) build a view-conditioned novel view diffusion model in the latent space and demonstrate its utility in sparse view 3D reconstruction. Chan et al. (2023) improve the view consistency of the diffusion model by reprojecting the latent features of the input view prior to diffusion denoising. More recently, Liu et al. (2023) propose a variant of this technique for performing text-to-3D with the generative ability of novel views enabled by fine-tuning language-guided priors on renderings from 3D datasets.

Generally, models trained in these methods are unable to accurately capture the view specifications, resulting in the generation of multiple views that are only *approximately* 3D consistent. Although our AGP also takes as input camera specifications to be viewpoint-aware, its purpose is merely to generate coarse geometries that will evolve subsequently into a 3D consistent object. Furthermore, aside from the differences in the formulation of these methods and ours on the text-to-3D task, our approach diverges significantly in that it does not capitalize on the appearance information, i.e., synthetic renderings, residing in 3D datasets, which is at the risk of compromising the visual priors learned in pre-trained diffusion models and thus may result in degraded visual quality.

## 3   METHOD

As aforementioned, the issues of multi-view or 3D inconsistency can be categorized from two perspectives: geometric inconsistency issues, which pertain to misplaced geometric structures in 3D, and appearance inconsistency issues, which relate to incorrect visual appearance modelling on the 3D geometric structures. Realizing that geometric inconsistency is the main reason for most 3D inconsistent results, our goal is to equip the 2D priors with the capability to produce 3D consistent geometric structures while retaining their generalizability. As a result, the generated consistent geometric structures play a continuous role in contributing to the modeling of intricate geometric details and visual appearance in pipelines that perform text-to-3D.

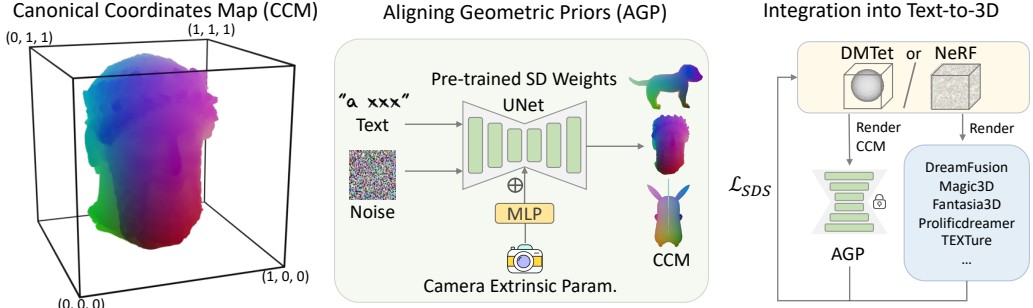

Figure 2: Method overview. We fine-tune the 2D diffusion model (middle) to generate viewpoint-conditioned canonical coordinates maps, which are rendered from canonically oriented 3D assets (left), thereby aligning the geometric priors in the 2D diffusion. The aligned geometric priors can then be seamlessly integrated into existing text-to-3D pipelines to confer 3D consistency (right), while retaining their generalizability to obtain high-fidelity and highly varied 3D content.

To this end, we propose to ensure the geometric priors in 2D diffusion act in a way that aligns with well-defined 3D geometries as depicted in 3D dataset (Section 3.1). Specifically, we assume to have access to a 3D dataset, which comprises extensive and diverse 3D models that are canonically oriented and normalized. We then render depth maps from random views, and convert them into canonical coordinates maps. Note we only render from rather coarse geometries, as the goal is merely to use 3D data for aligning rather than for generating geometric details. The benefits of using such 3D data are two-fold: i) all geometries are well-defined in 3D, so there is no ambiguity in their spatial arrangement; ii) by further injecting the viewpoint into the model, we can confer viewpoint awareness and eventually 3D awareness. Then, we fine-tune the 2D diffusion model to generate the canonical coordinates map under a specified view, eventually aligning the geometric priors in 2D diffusion. Finally, the aligned geometric priors can be seamlessly integrated into various text-to-3D pipelines (Section 3.2), significantly mitigating the inconsistency issues, resulting in the generation of high-quality and diverse 3D content. Figure 2 presents an overview.

## 3.1 Alignning geometric priors in 2D diffusion

Then, we elaborate technical details involved in aligning the geometric priors in 2D diffusion with well-defined 3D geometries during the lifting, while retaining its generalizability.

**Canonical Coordinates Map (CCM)**  To eliminate the distraction caused by the gauge free-dom and thereby ease the modeling, we assume that all objects within the same category adhere to a canonical orientation in the training data, a common practice in various publicly accessible datasets (Deitke et al., 2023; Chang et al., 2015). Note that, while the object orientation is assumed to be canonicalized per category, our objective is not to learn category-specific data priors. Instead, our aim is to extract general knowledge from a variety of objects in the 3D datasets, which will aid in aligning the 2D geometric priors. Analogous to (Wang et al., 2019; Shotton et al., 2013), the canonical object space is defined as a 3D space contained within a unit cube $\{x, y, z\} \in [0, 1]$. Specifically, given an object, we normalize its size by uniformly scaling the object such that the max extent of its tight bounding box has a length of 1 and is centered at the origin. While we can render coordinates maps at random views from these canonically oriented and uniformly normalized objects for training, we further propose to anisotropically scale the three components in the coordinate maps rendered from an object, such that the value of each component is within the range from 0 to 1. This anisotropic normalization amplifies the discrepancy of spatial coordinates on thin structures at different views, easing the perception of the 3D structures and thereby improving the 3D-awareness in the subsequent learning.

**Camera Condition**  Although the canonical coordinates maps contain rough viewpoint information, we found that the diffusion model has difficulties in exploiting it. Therefore, we inject the camera information into the model to improve viewpoint-awareness, following MVDream (Shi et al.,

2023). Specifically, we explicitly input the corresponding camera extrinsic parameters to the diffusion model, which is passed through an MLP before being fed to the middle layer of the diffusion model. Note that, in contrast to other models that rely on accurate viewpoint-awareness for generating consistent 3D, the use of camera conditions in our model is only to roughly generate coarse geometries that will evolve subsequently into a 3D consistent object.

**Fine-tuning 2D Diffusion for Alignment**   Given the pairs of the canonical coordinates map and its corresponding camera specification, we keep the architecture of the 2D diffusion model while slightly adapting it to be conditioned on camera extrinsic parameters. This enables us to leverage the pre-trained 2D diffusion model for transfer learning, thereby inheriting their generalizability in terms of highly varied subjects unseen in the 3D dataset. Finally, we fine-tune the diffusion model, originally intended for generating raw RGB or latent images, to generate the canonical coordinates map under a viewpoint condition, eventually aligning the geometric priors in 2D diffusion.

**Implementation Details**   By default, we conduct experiments based on the Stable Diffusion model (we use v2.1), which is a commonly used public large pre-trained text-to-image diffusion model.

*3D dataset.* We use a public 3D dataset – Objaverse (Deitke et al., 2023), which contains around 800k models created by artists, to generate the data for fine-tuning.

By having a significant portion of the 3D objects in canonical orientation and only a few remaining misoriented, we are able to achieve satisfactory outcomes without the necessity of manually correcting those misoriented ones. On the other hand, due to the presence of considerable noise in the textual annotations, we employ a 3D captioning model (Luo et al., 2023) to augment the textual description of each 3D asset and randomly switch between the augmented caption and its original textual annotation (typically names and tags) during the training. In addition, to ensure relevance, we apply a filtering process based on tags to eliminate 3D assets such as point clouds and low poly models, resulting in approximately 270k objects.

*Camera sampling.* We render canonical coordinates maps from the 3D objects. The camera is randomly positioned at a distance ranging from 0.9 to 1.1 units, with a field of view set at 45 degrees. Additionally, the camera's elevation is randomly varied between -10° and 45°. As we do not rely on visual appearance information, we are able to utilize a fast rasterization renderer for generating the training data, avoiding the computational intensity associated with ray tracing renderers.

*Training.* We fine-tune our model on the latent space of Stable Diffusion using Diffusers (von Platen et al., 2022). Note that the canonical coordinates map is directly treated as a latent image for the latent diffusion model to produce. This leads to an attractive feature that our aligned geometric priors can be trained fast, without involving the encoding and decoding process of the VAE. We keep the default optimizer setting, as well as the $\epsilon$-prediction. Since we input the camera extrinsic parameters as a condition to the diffusion model, the training objective is now formulated as follows:

$$\mathcal{L}_{LDM} := \mathbb{E}_{c,y,z,t,\epsilon \in \sim N(0,1)}\left[||\epsilon - \epsilon_\theta(c, \tau_\theta(y), z_t, t)||_2^2\right], \text{ where c is the camera extrinsic parameters,}$$

$y$ the input text prompt and $\tau_\theta(y)$ its embedded feature using tokenizer, and $z_t$ the noisy latent image generated by adding noise $\epsilon$ to a clean latent image $z$ at a diffusion timestep $t$.

## 3.2 INTEGRATION INTO TEXT-TO-3D

Finally, we elaborate on how to integrate our aligned geometric priors into existing pipelines using different 3D representations, significantly mitigating their inconsistency issues and achieving state-of-the-art text-to-3D performance. To showcase such compatibility, we provide demonstrations of two state-of-the-art text-to-3D methods that utilize different 3D representations, namely, Fantasia3D (Chen et al., 2023), which explicitly disentangles the geometry and appearance modeling and uses a hybrid representation – DMTet (Shen et al., 2021) – for the underlying 3D geometry, and DreamFusion (Poole et al., 2022), which employs the neural radiance field (NeRF) (Mildenhall et al., 2020) as the 3D representation. Please see Figure 3 for the system pipelines of integrating our aligned geometric priors into these two methods. For more technical details, we refer readers to the original papers, as we do not elaborate on them further in this context.

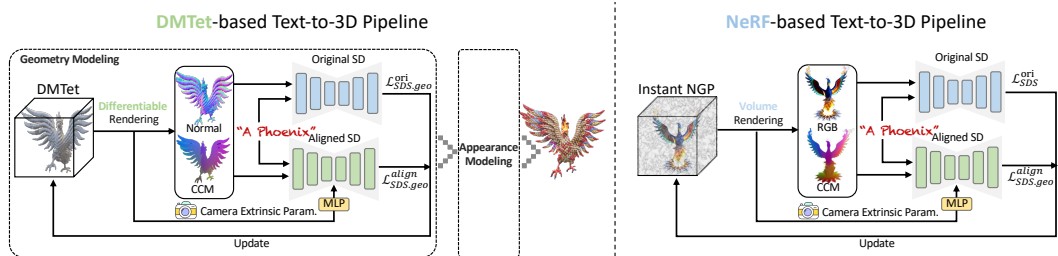

Figure 3: Seamless integration of our AGP in various text-to-3D pipelines.

**DMTet-based Pipeline** For the sake of clarity, we refer to the variant obtained by integrating our aligned geometric priors into Fantasia3D as the DMTet-based pipeline. All that is required is an additional parallel branch, to incorporate our aligned geometric priors for supervising the geometry modeling in the original pipeline. With this seamless integration of our aligned geometric prior, high-quality and view-consistent results can be easily achieved, without the need for carefully designed initialization shapes as in the original pipeline.

*Optimization.* Specifically, we add this additional supervision imposed by our aligned geometric priors in both the coarse and fine geometry modeling stages. Simply, our aligned diffusion model takes as input the canonical coordinates map and produces the SDS loss to update the 3D representation. Then, the final loss function in the geometry modeling can be written as $\mathcal{L}_{SDS \cdot geo} = \lambda^{ori} \mathcal{L}_{SDS \cdot geo}^{ori} + \lambda^{align} \mathcal{L}_{SDS \cdot geo}^{align}$, where the first term is the geometry SDS loss derived from the original diffusion model, the latter is the SDS loss derived from our aligned geometric priors. Here, $\lambda^{ori}$ and $\lambda^{align}$ are the weights to balance their effects. The revised system pipeline is shown on the left of Figure 3. Note that this integration is implemented only in the coarse and fine geometry stages, while the appearance modeling stage is untouched.

**NeRF-based Pipeline** NeRF is another common choice for the 3D representation in text-to-3D, as it is more friendly for optimization compared to traditional discrete meshes, and can also be combined with volume rendering for great photo-realism. Specifically, we base on a popular implementation (Guo et al., 2023) of the pioneer – DreamFusion (Poole et al., 2022), which uses NeRF as the 3D representation, and refer to it as the NeRF-based pipeline. Particularly, the 3D scene is represented by Instant-NGP with an extra MLP for modeling the environment map, allowing the modeling of rich details with low computing cost. Then we can volume-render the 3D object/scene to obtain the RGB images and feed them into the Stable Diffusion to calculate SDS loss.

*Optimization.* During the lifting optimization, we render the canonical coordinates map and feed it to our aligned geometric priors to calculate the geometry SDS loss $\mathcal{L}_{SDS \cdot geo}^{align}$ to help update the geometry branch of the NeRF, in addition to the origin SDS loss $\mathcal{L}_{SDS}$ calculated with the RGB image. Similar to the previous integration, the final loss is the weighted combination of the original SDS loss and our aligned geometric SDS loss: $\mathcal{L}_{SDS} = \lambda^{ori} \mathcal{L}_{SDS}^{ori} + \lambda^{align} \mathcal{L}_{SDS \cdot geo}^{align}$, where $\lambda^{ori}$ and $\lambda^{align}$ are the weights balancing these two terms. Note our AGP continues to model 3D consistent coarse geometries in this pipeline, while again leaving the appearance modeling untouched.

## 4 TEXT-TO-3D GENERATION

We present the qualitative and quantitative evaluation of the text-to-3D pipelines as described in Section 3.2, as well as comparison results against other text-to-3D baseline methods. For convenience and clarity, we refer to the DMTet-based pipeline and NeRF-based pipeline as *Ours (DMTet-based)* and *Ours (NeRF-based)*, respectively. Furthermore, depending on the different pre-trained diffusion models used in its *original* pipeline, we developed two versions of Ours (NeRF-based), namely, *Ours (NeRF-based IF)* using Deepfloyd IF, and *Ours (NeRF-based full)* using DeeFloyd IF first and then Stable Diffusion. Please refer to the supplementary for more details.

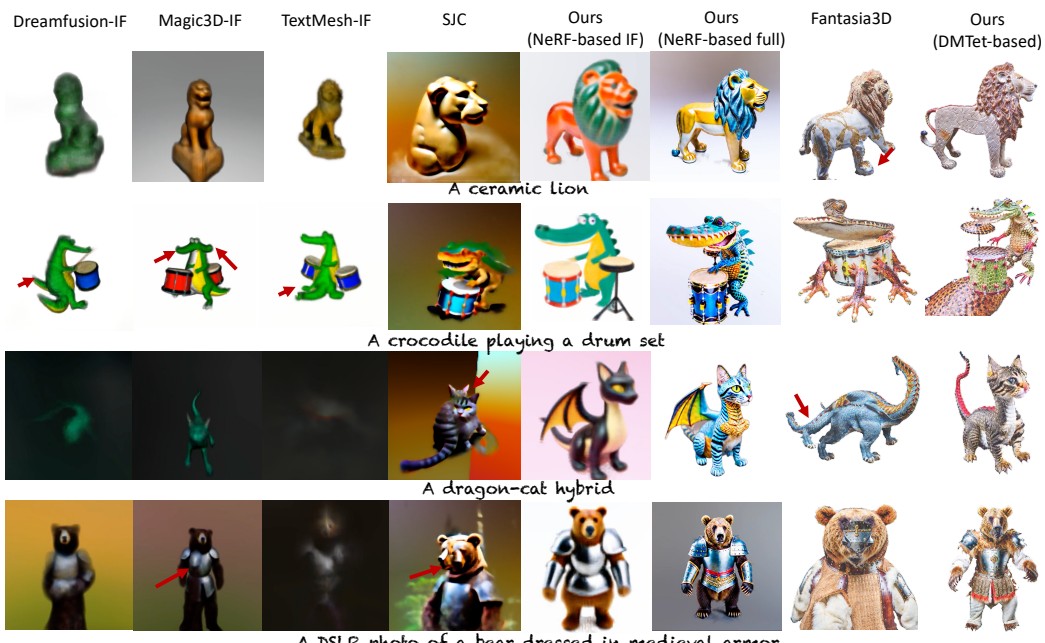

Figure 4: Visual comparisons. Compared to other competing methods, our text-to-3D pipelines can generate high-fidelity 3D content with high 3D consistency. See 3D inconsistency issues in the baseline results (highlighted by red arrows).

**Baselines**  We extensively compare to baselines as follows: i) *Fantasia3D*, based on which our DMTet-based pipeline is implemented. We compare our DMTet-based pipeline against it to show specifically the effectiveness of our AGP; ii) *DreamFusion-IF*, which replaces the unreleased Imagen (Saharia et al., 2022a) with – DeepFloyd IF (IF, 2023). We compare our NeRF-based against it to validate our AGP again; And more other baselines implemented in (Guo et al., 2023), including iii) *SJC* (Wang et al., 2023a), that applies the chain rule on the learned gradients of a diffusion model and backpropagates the score of a diffusion model through the Jacobian of a differentiable renderer to optimize a 3D world; iv) *Magic3D-IF* (Lin et al., 2023) is a hybrid pipeline that uses NeRF at the coarse stage and then converts it into a mesh for fine details. We also adapt it to use DeepFloyd IF for SDS calculation; v) *TextMesh-IF* (Tsalicoglou et al., 2023a), which is also a DeepFloyd IF-based implementation, is similar to Magic3D but uses an SDF-based representation at the coarse stage. vi) *MVDream* (Shi et al., 2023), which is a concurrent work to us. Note that, since their official implementation is unavailable by the time of our submission, we use the same prompts as listed on their website for side-by-side comparisons.

Due to various reasons, we were not able to obtain the original implementation of most baselines. Therefore, except for Fantasia3D and MVDream, we use the implementation from Guo et al. (2023) for all baselines. We consider these implementations to be the most reliable and comprehensive open-source option available in the field. By default, we use the Stable Diffusion model as the prior, except those with the name suffixed "IF" use DeepFloyd IF within the pipeline.

**Quantitative Evaluation**  It is important to acknowledge that currently, there is a lack of well-established metrics that can quantitatively and comprehensively evaluate the text-to-3D results from various perspectives. In this work, our primary focus lies in generating multi-view consistent 3D content, rather than placing specific emphasis on enhancing the appearance or texture quality of existing pipelines. So we focus on quantitatively evaluating the multi-view consistency of the 3D results. Specifically, we randomly select 80 text prompts from the DreamFusion gallery (dre, 2023), and perform text-to-3D synthesis to generate 80 results using each method. We then manually check and count the number of occurrences of 3D inconsistencies (e.g., multiple heads, hands, or legs) and report the success rate, i.e., the number of 3D consistent objects divided by the total number of generated results. Note that, while the 3D consistency can be vague sometimes to describe, the

human participants involved are researchers with highly relevant backgrounds such as Computer Vision and Computer Graphics, so they roughly follow a general definition of 3D consistency in the evaluation. As shown in Table 1, our method outperforms other methods by a large margin. Our success rates are over 85+% in both pipelines, while the previous methods are only around 30%.

| | Dreamfusion -IF | Magic3D -IF | TextMesh -IF | SJC | Fantasia3D | Ours (DMTet-based) | Ours (NeRF-based IF) |
|---|---|---|---|---|---|---|---|
| Cons. Rate ↑ | 30.0% | 35.0% | 23.8% | 7.5% | 32.5% | **87.5%** | **88.8%** |

Table 1: Quantitative comparison results for the 3D consistency rate.

**Qualitative Evaluation** As shown in Figure 4, By integrating our AGP into Fantasia3D, i.e., *Ours (DMTet-based)*, the results have been significantly improved. The original Fantasia3D only produced coarse and inaccurate results without hand-crafted initial geometries. We believe this is due to the domain gap between the rendered normal map and the geometric information extracted in the latent space of the Stable Diffusion, resulting in optimization difficulties in converging to a reasonable 3D shape. As for Ours (NeRF-based), the generated results clearly have high 3D consistency and possess a more realistic appearance. This is because our aligned geometric priors only contribute to the geometry modeling during the lifting, while they do not compromise the appearance modeling guided by the powerful visual priors learned by Stable Diffusion from billions of real images. In general, the 3D results generated by most remaining baselines, even when equipped with the more powerful Deepfloyd IF, suffer from multi-view inconsistency (easier to see from spinning views in the supplementary file). Note that, when evaluating the results, we focus on assessing the 3D consistency, and hence do not heavily penalize blurry images, as they can be caused by the use of DeepFloyd IF with limited computing resources. Last, although the concurrent work, MVDream, can also resolve the multi-view inconsistency problem, we observe that it is prone to overfit the limited 3D data, consequently resulting in a compromise of the generalizability in the original powerful 2D diffusion model. Since they use synthetic multi-view renderings for fine-tuning their multi-view diffusion model, the appearance of the generated results lacks the desired level of photo-realism.

**User Study** We also conducted a user study on 30 generated 3D results of relevant methods. Each participant was presented with videos rendered from the 3D models obtained by various methods based on one of the 30 text prompts. Then, they were requested to choose a 3D model they favored the most, considering only the 3D consistency. We report the rate of preference for each method in the inset pie chart. As shown, while this evaluation is more noisy, yet our method consistently outperforms the competing methods by a large margin, showing the robustness of our method in generating results of high 3D consistency.

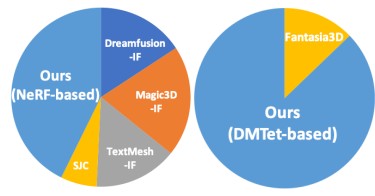

## 5 CONCLUSION

We introduced Aligned Geometric Priors (AGP), which is obtained by fine-tuning a pre-trained 2D diffusion model to generate viewpoint-conditioned coarse geometric maps of canonically oriented objects, thereby conferring 3D awareness. AGP is generic and can be seamlessly integrated into various existing pipelines to generate 3D objects of high consistency. Most importantly, AGP improves the geometry modeling and does not compromise the appearance modeling guided by strong priors learned from billions of real images. While AGP has shown state-of-the-art performance in text-to-3D, we also note a few limitations. Our work does not directly consider appearance modeling, where inconsistency may still arise *rarely* due to the remaining ambiguity in the mapping from the geometric structure to its associated appearance. Early in our development, we attempted to incorporate an appearance generator by fine-tuning the 2D diffusion model to generate the appearance image conditioned on a given canonical coordinates map. Unfortunately, this approach resulted in overfitting to the renderings derived from 3D data, leading to 3D results that lacked the desired level of photorealism. We leave the study in this direction for future work. All that being said, we believe the work opens up a novel "less is more" direction of utilizing relatively limited 3D data to enhance 2D diffusion priors for text-to-3D synthesis.

# 6  ACKNOWLEDGMENTS

We would like to thank the anonymous reviewers for their constructive comments. We thank Jianxiong Pan for his effort in data preparation. Weiyu Li was partly supported by the Shenzhen Collaborative Innovation Program (CJGJZD20210480926010 03).

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

## A  APPENDIX

### A.1  ABLATION STUDY

We also validate the effectiveness of key algorithmic design choices using the NeRF-based pipeline. Regarding the geometric representation, we have also derived variants of our method using two types of geometric representations, namely depth and normal maps, and show that, while these two geometric representations can also be incorporated into our pipeline for aligning the geometric priors in 2D diffusion, they indeed fail in producing 3D results as high-quality as CCM.

On one hand, a depth map is a single-channel map storing the per-pixel distance between the surface to the camera. Compared to CCM, the depth map alone does not contain essential information about the relative orientation of the 3D object and the camera, so the network can solely rely on the camera information for learning 3D-aware priors, consequently leading to degraded robustness of the learned 3D priors. In stark contrast, the CCM alone has actually encoded the information of the orientation to some extent. This is manifested by the distinct color coding results of CCMs from different viewpoints. As a result, the variant using depth maps is more prone to multi-view inconsistent issues, compared to our method using CCM. On the other hand, a normal map stores the local directions of fragments on the 3D surface, we use normal maps obtained from the canonical coordinate space for fair comparisons. Our experimental results show that the text-to-3D optimization process has difficulty in generating a 3D world of which the derivative follows faithfully generated normal maps, and may fail to generate meaningful 3D results. Those successfully generated 3D results tend to be slightly smoother regarding the geometric and appearance details of the surface.

Last, we have also conducted a study where the camera information injection is removed from our method. The results demonstrate that removing camera parameters leads to a decrease of the consistency in the generated results, due to the lack of critical camera information. Please see Figure 5 for the visual results and refer to Section A.6 for more.

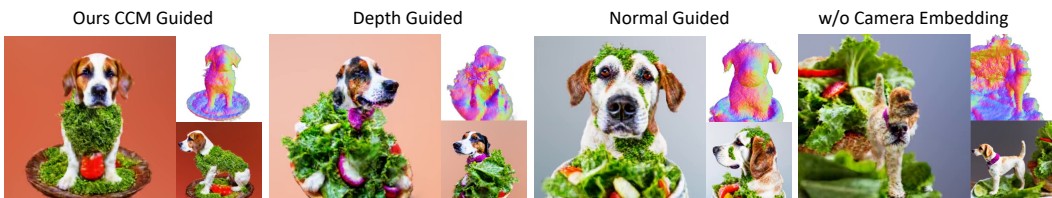

Figure 5: Ablation study. Replacing CCM with depth maps results in inconsistent geometry and the use of normal maps as guidance produces smooth geometry. Additionally, removing the camera embeddings led to noisy and inconsistent results.

### A.2  MORE IMPLEMENTATION DETAILS

**Dataset**  Notably, while there is no explicit specification of the coordinate system in Objaverse, most 3D objects in Objaverse are uploaded by artists who usually adhere to a convention regarding the orientation when creating 3D assets. Furthermore, for some special categories, such as characters, we filter out misoriented data simply by the ratio of its axis-aligned bounding box. After these filterings, we found that approximately 80% (based on statistical random samplings) of the remaining data are orientated canonically, which is sufficient for our purpose as evidenced by extensive results in our paper (please also refer to the appendix for more discussions on the noise in the dataset).

**Fine-tuning**  We use Diffusers (von Platen et al., 2022) to finetune the Stable Diffusion v2.1 using our rendered CCMs at a resolution of 64 x 64. During the fine-tuning process, we remove the encoder of the VAE and directly concatenate the CCM with corresponding alpha channels as the latent to the UNet, with the background color of the CCM set to random. We use the default parameters as in Diffusers, including setting the learning rate to 1e-5 with the constant scheduler, and a

batch size of 96 per GPU with 4 gradient accumulation steps. The entire fine-tuning process takes approximately 2 days using 8 V100 GPUs for 100k steps.

**Ours (DMTet-based)**   We integrate our Aligned Geometric Priors in the official repository of Fantasica3D (Chen et al., 2023) as described in Section 3.2. We follow the same parameters as in the original paper. We also disentangle the learning of geometry and appearance. It takes about 12 and 8 minutes to generate a fine geometry and its corresponding Physically-Based Rendering (PBR) materials, respectively, for each object. For the time step range of SDS loss, We adopt a uniform sampling strategy of annealing from [0.5, 0.98] to [0.05, 0.5]. The whole process takes about 0.5 hours to generate each object using 4 V100 GPUs.

**Ours (NeRF-based full)**   We implement it in the threestudio (Guo et al., 2023), which implemented a diverse set of state-of-the-art text-to-3D generation pipelines. Specifically, we use Instant-NGP (Müller et al., 2022) as the 3D representation to optimize, which uses a multi-resolution hashgrid to predict the RGB and the density of the sampled ray points. The sampled camera views follow the same protocol as the render dataset to fine-tune the UNet. We use DeepFolyd at the coarse stage with 64 x 64 resolution and then switch to Stable Diffusion with 512 x 512 for detailed optimization. In addition, we also use time annealing, negative prompts, and CFG rescaling tricks from open source implementation for improved performance. For SDS, the maximum time step is decreased from 0.98 to 0.5 linearly and the maximum time step is kept to 0.02. We use a rescale factor of 0.7 for the CFG rescale. The whole process takes about 1 hour to generate each object with 10, 000 steps using 2 V100 GPUs.

## A.3   MORE COMPARISON RESULTS USING PROMPTS FROM MVDREAM

Note that, since MVDream's official implementation is unavailable by the time of our submission, we use the same prompts as listed on their website for side-by-side comparisons. We present the visual comparisons in Figure 6. Although the concurrent work, MVDream, can also resolve the multi-view inconsistency problem, we observe that it is prone to overfit the limited 3D data, consequently resulting in a compromise of the generalizability in the original powerful 2D diffusion model. Specifically, as shown in the results, MVDream misses the "backpack" in its generated result presented with the prompt "an image of a pig carrying a backpack". Additionally, since they use synthetic multi-view renderings for fine-tuning their multi-view diffusion model, the appearance of the generated results lacks the desired level of photorealism.

## A.4   GENERALIZABILITY

Our method can effectively address the notorious multi-view inconsistency problem, and equally importantly, retains to the maximum extent the generalizability of the foundation text-to-image model in terms of the highly varied appearance and geometric details. We would like to highlight that the pre-trained text-to-image diffusion model is powerful, and this preservation of its generalizability is particularly attractive, as it can lead to more diverse and highly realistic 3D generation results. We have achieved this by only aligning the geometric priors in 2D diffusion using the coarse geometric information (CCM) of well-defined geometries, without compromising the original diffusion priors in the text-to-3D pipeline regarding detailed appearances and geometries. This is in contrast to other models that hinge on all appearance and geometric details in the 3D dataset for fine-tuning the diffusion priors, which is at the risk of compromising the integrity of the original geometric priors learned in the pre-trained text-to-image foundation model, leading to the degradation of the generalizability in terms of highly diverse and photo-realistic 3D objects. While we have validated this through the comparison results against MVDream (Figure 6), we have also prepared more extensive qualitative results to further showcase such ability of our method to generalize smoothly to 3D worlds with highly realistic and varied appearance and geometric details, that are unseen in the 3D dataset. See Figure 7 for more visual results.

More specifically, the generalization encompasses both geometric and appearance detils. In our work, we only perform fine-tuning using the coarse geometric structures, which enables us to preserve the highly realistic appearance details of the original image priors learned from a large real-world dataset. Interestingly, we found that in our result of using the prompt "corgi riding on a rocket", the rough geometry obtained with only AGP priors is not yet completely accurate. How-

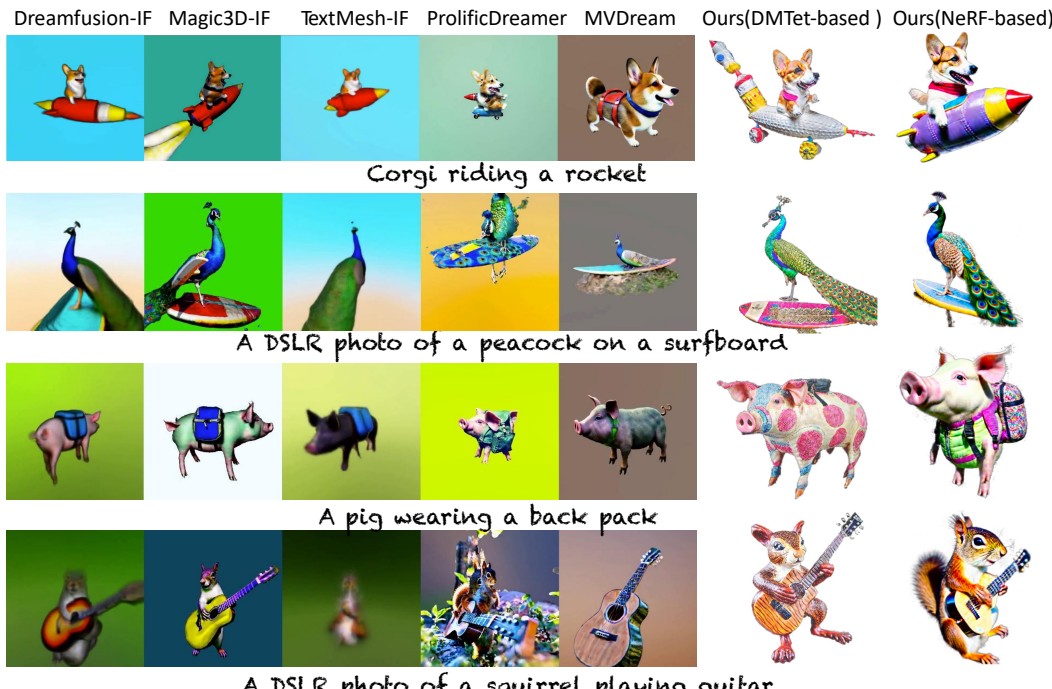

Figure 6: Side-by-side visual comparisons using prompts from MVDream. Note that some key concepts in the prompts are missing in MVDream results, such as the rocket, backpack, and squirrel missing in their results.

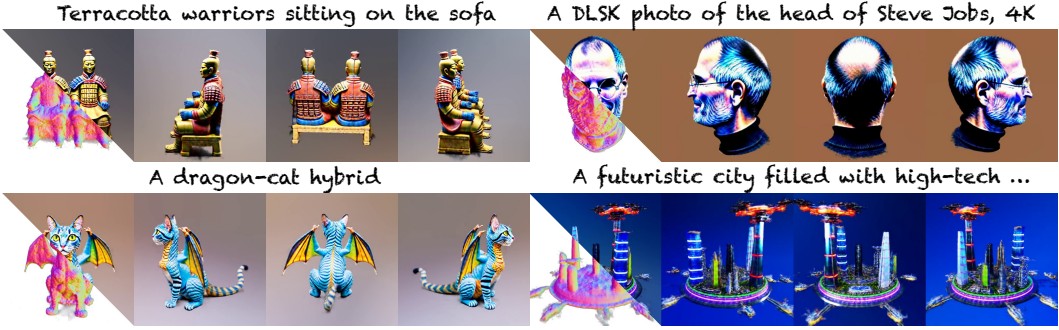

Figure 7: Our method can produce a wide range of 3D results spanning from weird characters to fancy architectural models. It is worth noting that none of these 3D models exists in the training data. Our approach can generalize very well to these highly diverse and high-quality results.

ever, after optimizing with SDS of image priors, we are able to fix the slightly incorrect geometry and obtain more detailed geometric and appearance details, as shown in Figure 8.

## A.5 THE EFFECT OF DIFFERENT SETTINGS OF DATASET

We used a set of rules to filter the original dataset and obtained approximately 270k objects, of which approximately 80% have a consistent orientation (estimated based on statistical random samplings). In this section, we aim to investigate the impact of a different-sized dataset on the performance of our proposed method.

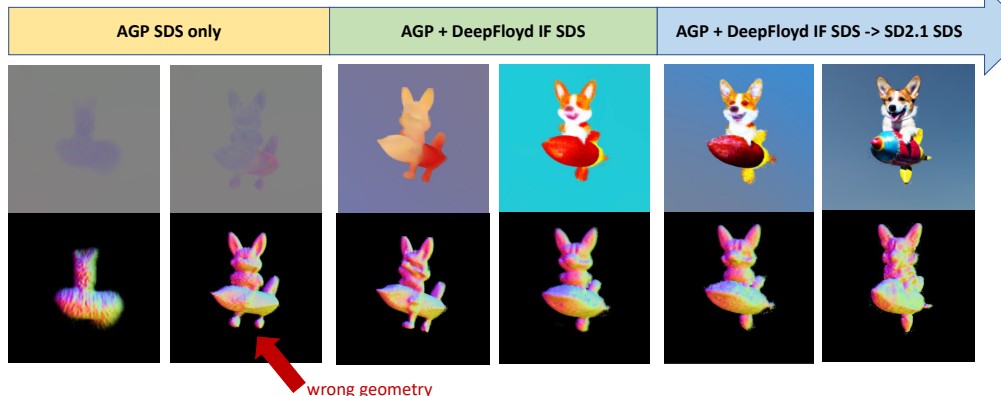

Figure 8: Visualization of each stage of our NeRF-based method of prompt "Corgi riding a rocket". Using only AGP SDS as supervision may result in incorrect geometry. However, the image priors can help eliminate these errors in the later stages, thereby improving the generalization performance.

We calculated the distance between the front-view images and their corresponding captions' blip feature and sorted them accordingly. We then manually screened the top 20k data, resulting in a dataset of 18,488 instances with a uniform orientation and high correspondence with their descriptions.

From the experimental results obtained by training on such a smaller dataset, we are surprised to find that even with a smaller dataset of only 20k instances for fine-tuning, our text-to-3D pipeline still exhibits strong generalization and is capable of generating diverse 3D objects. As shown in Figure 9, the majority of the generated results are satisfactory, with some lacking intricate details. We hypothesize that this may be attributed to the inadequate acquisition of 3D prior knowledge from a rather limited dataset, which consequently leads to the occurrence of coarse geometries. This observation necessitates the need for further exploration and a more thorough study of the impact of the 3D dataset.

## A.6 MORE DETAILS OF ABLATION STUDY

We conducted ablation experiments using a manually curated small dataset ( 20k data) for each object. We rendered CCM, normal, and depth maps for each object and used the same parameters to fine-tune the Stable Diffusion model. We then tested using the same prompt and seed. Extensive visual results are presented in Figure 10.

## A.7 QUANTITATIVE EVALUATION OF THE APPEARANCE OF GENERATED OBJECTS

We also present quantitative results using appearance-related metrics. However, it is worth noting that the image diffusion models used in Dreamfusion and Magic3D are not open-sourced, and the performance of the versions used here may differ from the original papers. For fairness, we used the same image generation model, DeepFloyd IF (IF, 2023) for all methods, and NeRF as the 3D representation.

To assess the appearance quality, we rely on DreamFusion's R-Precision metric. When provided with rendered images, R-Precision gauges the top-N accuracy in retrieving the correct caption from a pool of distractions, utilizing CLIP scores derived from averaging the similarity between each of four distinct rendered images and a caption. We employ the CLIP ViT-B/32 and ViT-B/16 models to respectively compute the top-1 and top-5 R-Precision, using the identical set of 80 prompts for evaluation as presented in the paper. Please note that since other methods may not have access to the original image diffusion model, the performance of reproduced code may vary. Therefore, the specific numerical results can only serve as a reference.

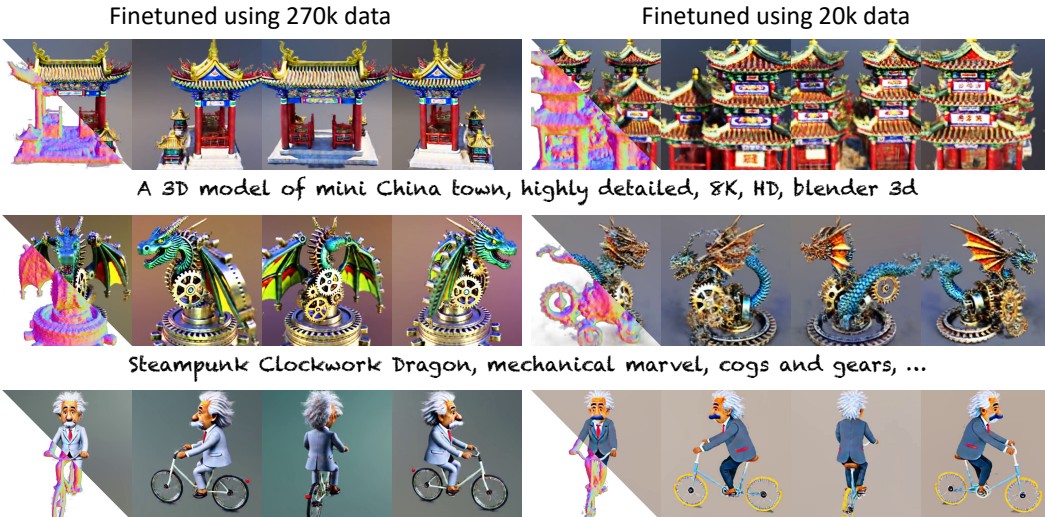

Finetuned using 270k data      Finetuned using 20k data

A 3D model of mini China town, highly detailed, 8K, HD, blender 3d

Steampunk Clockwork Dragon, mechanical marvel, cogs and gears, ...

Albert Einstein with grey suit is riding a bicycle

Figure 9: Comparison of finetuning results with different amounts of data. Despite reducing the training data to 1/10 of the original amount, the finetuned models still exhibit strong generalization capabilities (right). Note that, the model fine-tuned on fewer data tends to produce results of slightly fewer details and probably a bit more bulky geometries.

| Method | CLIP R-Precision(%) | | | |
| | CLIP B/32 | | CLIP B/16 | |
| | R@1 | R@5 | R@1 | R@5 |
|---|---|---|---|---|
| Magic3D | 60.1 | 71.5 | 62.7 | 77.7 |
| TextMesh | 51.7 | 65.1 | 55.1 | 78.4 |
| SJC | 40.2 | 51.2 | 52.5 | 62.5 |
| DreamFusion | 59.7 | 70.2 | 61.6 | 74.3 |
| Ours (NeRF-based) | **77.5** | **84.9** | **88.7** | **92.3** |

Table 2: Quantitative results demonstrating the coherence of visual appearance with their corresponding prompts, as assessed by CLIP retrieval models.

The results highlighted in Table 2 underscore the significant advantages of our proposed approach in terms of appearance. This superiority primarily stems from the fine-tuning process which only learns from coarse geometries and leaves the appearance model of the text-to-3D pipeline untouched.

## A.8 MORE DETAILS ABOUT THE USER STUDY

In contrast to the quantitative evaluation above, the human users involved in this study are from various loosely related backgrounds in CS and EE. The interactive interface used in the user study can be found in Figure 11.

## A.9 MORE TEXT-TO-3D RESULTS

We present more text-to-3D synthesis results obtained with our methods (Figure 12, Figure 13, Figure 14, and Figure 15).

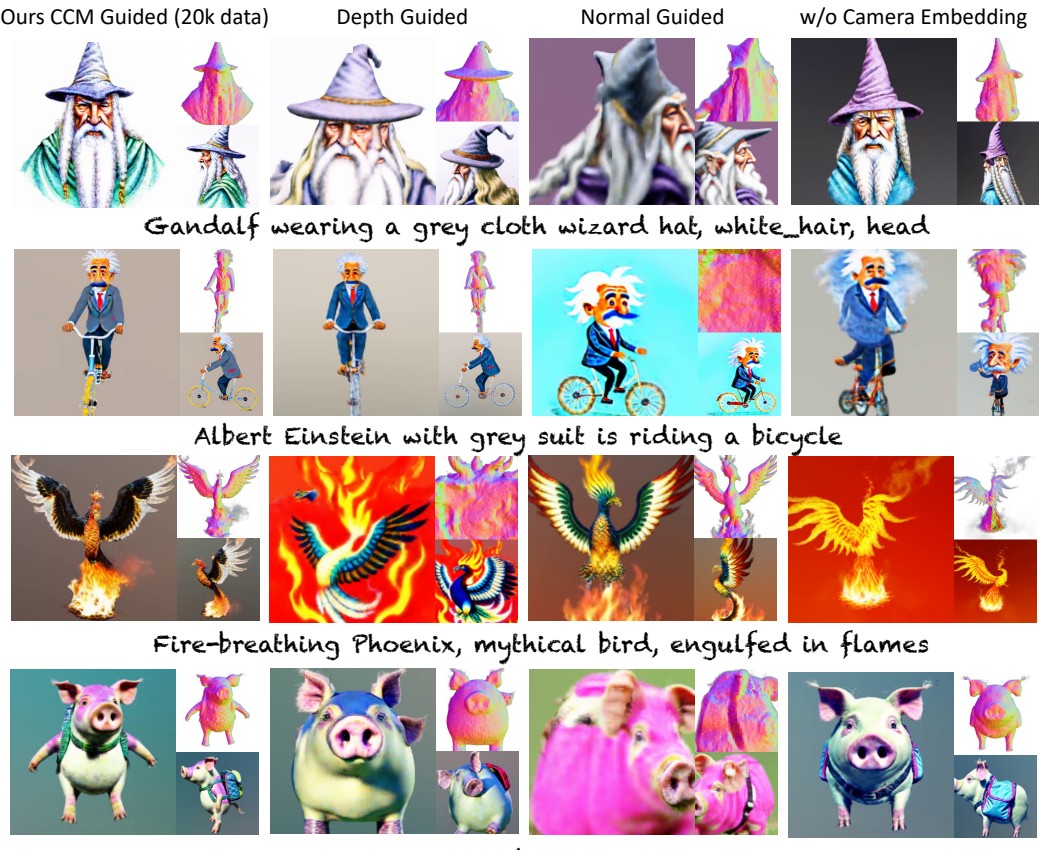

Figure 10: Comparing ablate settings with different guidance methods. Our proposed CCM proves superior to depth and normal guidance, delivering consistent and convincing geometry. Substituting CCM with depth maps leads to round (e.g. the pig) and inconsistent objects (e.g. the Gandalf and phoenix), while normal maps as guidance make the optimization unstable (e.g. the Gandalf, Einstein, and pig), leads to smoother (e.g. the phoenix) or noisy (e.g. the chihuahua) geometry. It is essential to note that removing the camera embedding may result in inconsistent results, particularly on complex structures (e.g. the Einstein).

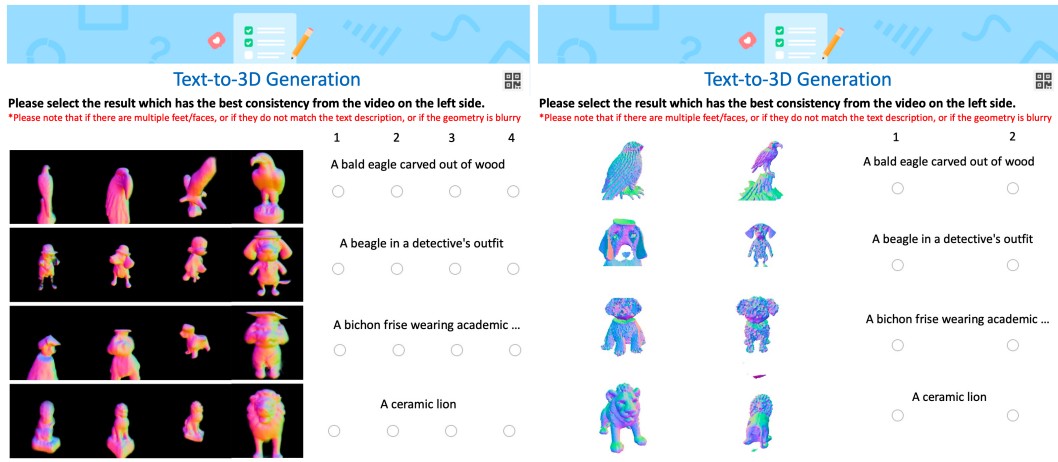

Figure 11: User study interface. We have designed two survey forms separately for the NeRF-based method and the DMTet-based method and provide users with multiple rendered videos of generated results. The user needs to select the one they consider to be of the best quality from all results.

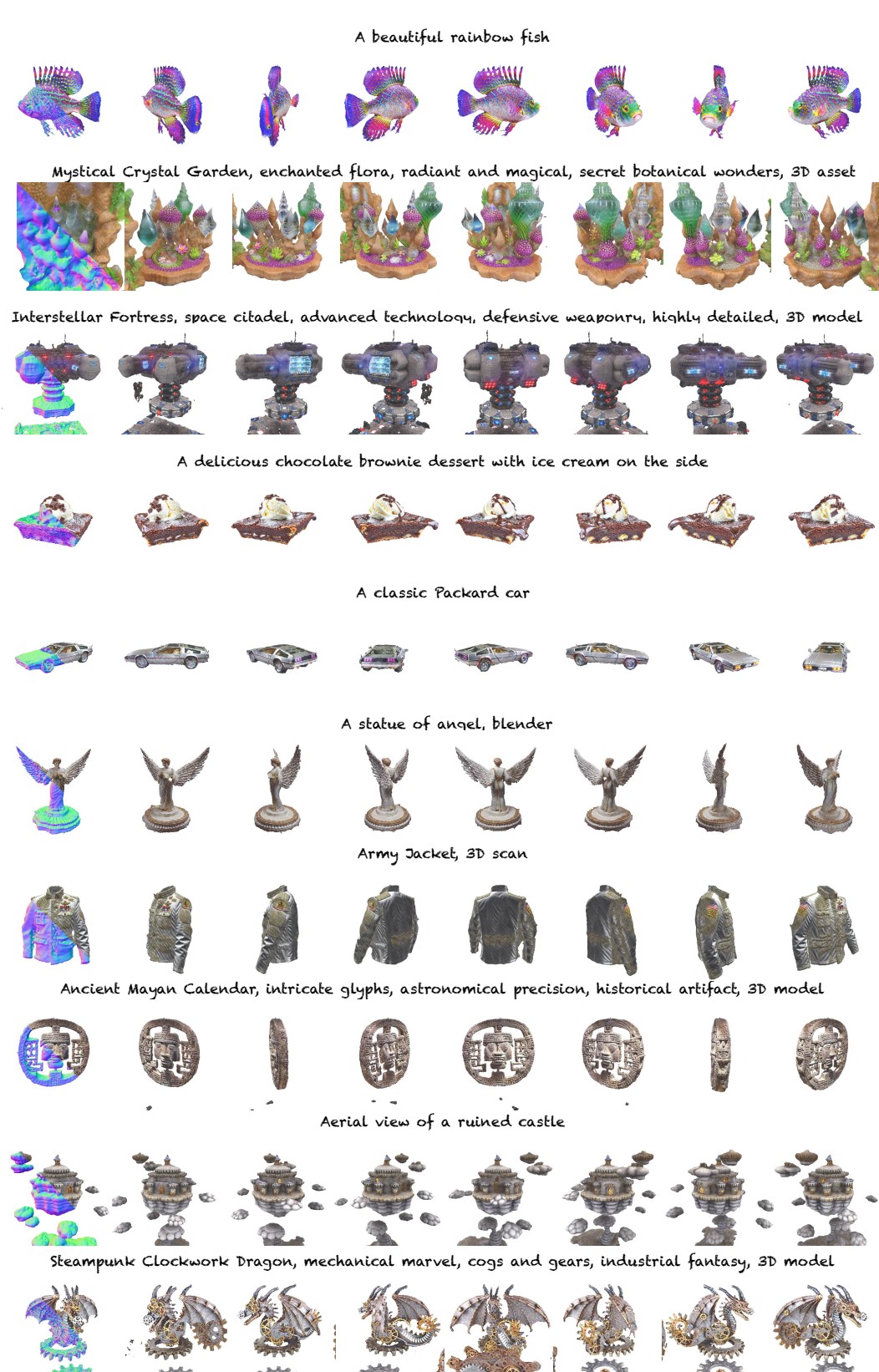

Figure 12: More generated results using our proposed DMTet-based model.

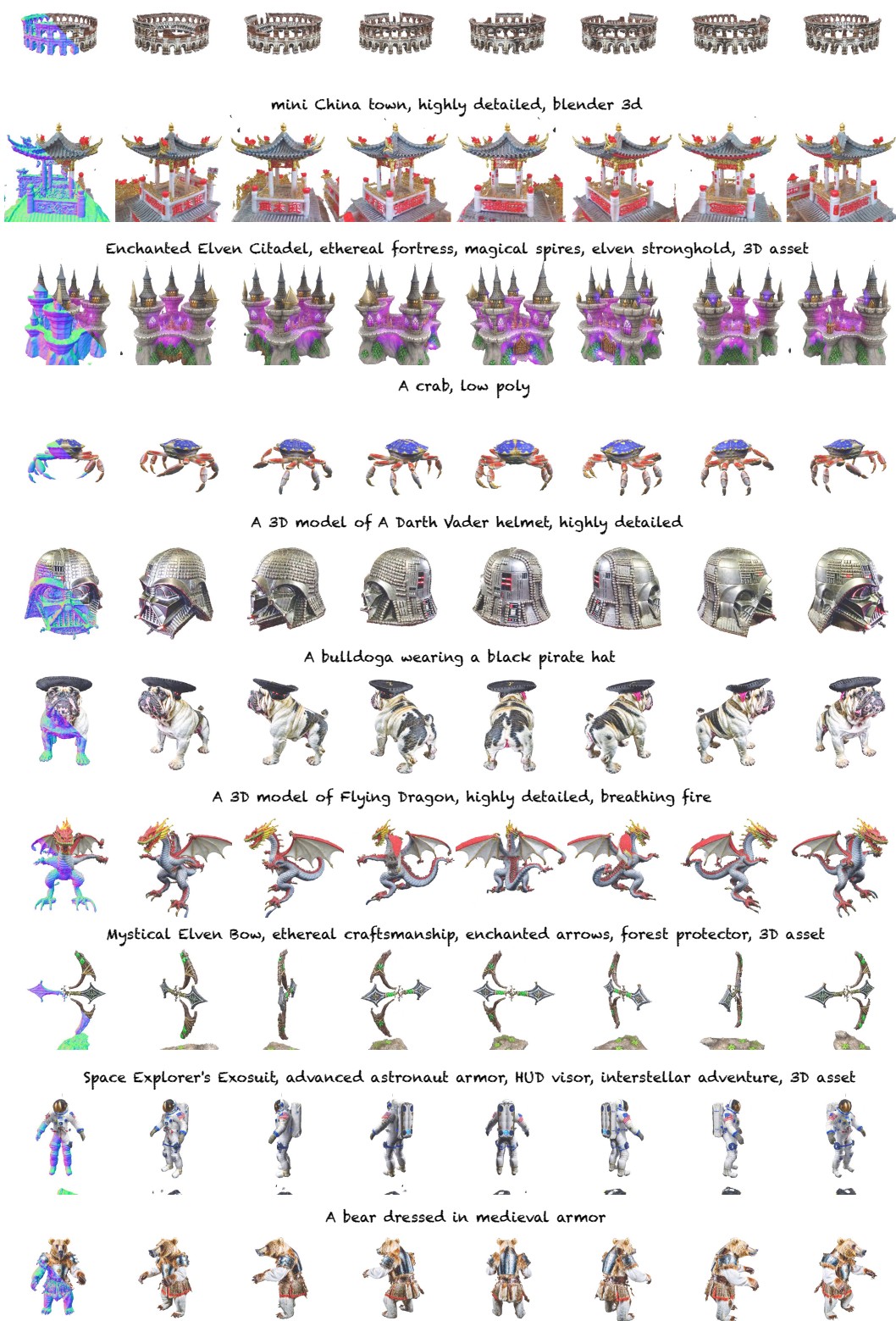

Figure 13: More generated results using our proposed DMTet-based model.

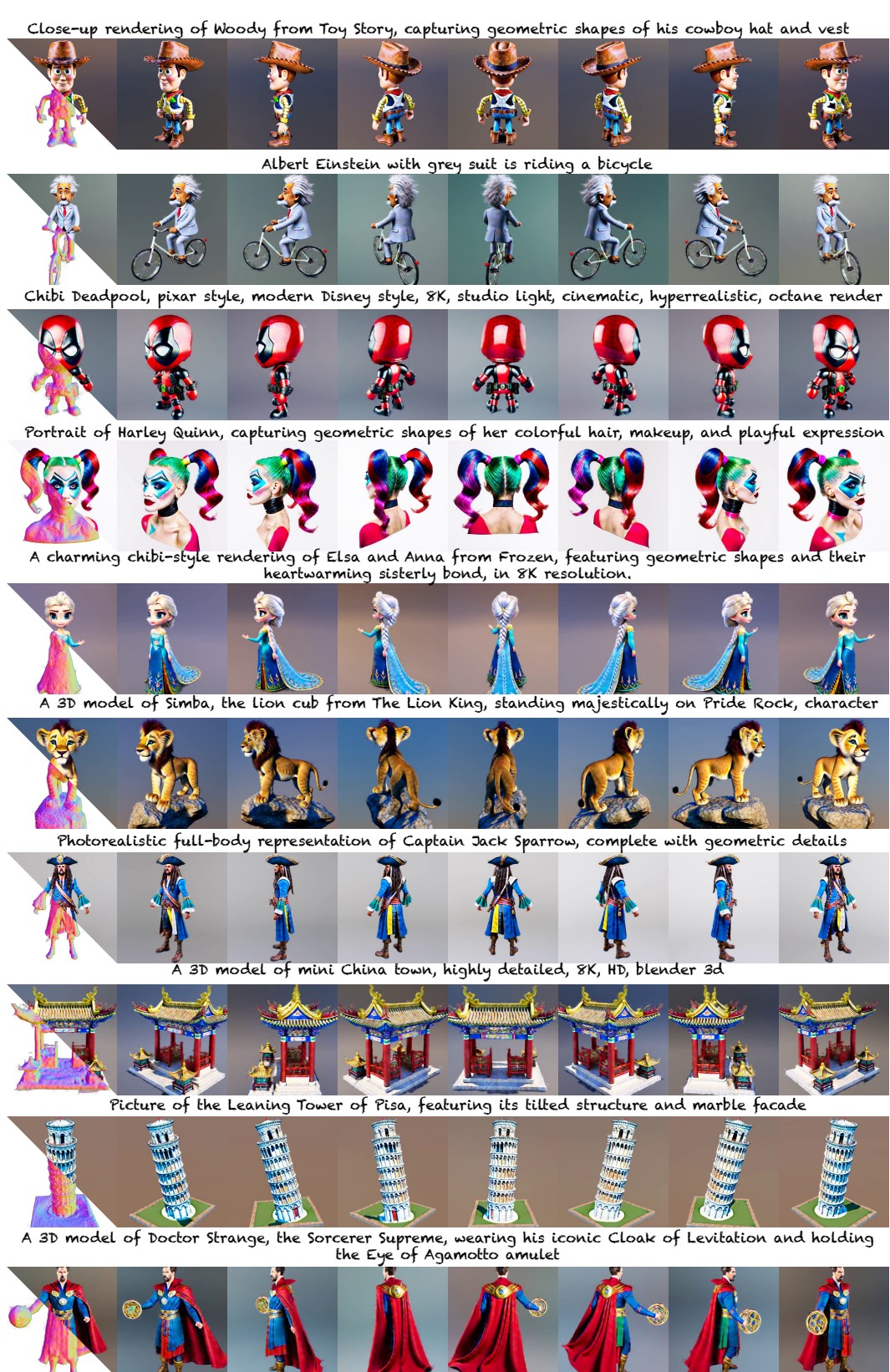

Figure 14: More generated results using our proposed NeRF-based model.

Figure 15: More generated results using our proposed NeRF-based model.

