# OpenReview forum: "SweetDreamer: Aligning Geometric Priors in 2D diffusion for Consistent Text-to-3D"
_ICLR.cc/2024/Conference — ICLR 2024 poster_

### Official Review · Reviewer_UgSj · 2023-10-30

**Soundness:** 3 good
**Presentation:** 3 good
**Contribution:** 2 fair
**Rating:** 5
**Confidence:** 4

**Summary:**

In this paper, the author present SweetDreamer, a text-to-3D generation model. They address the limitations of existing 2D diffusion models by aligning the 2D geometric priors with well-defined 3D shapes from a 3D dataset. To resolve the multi-view inconsistency, SweetDreamer incorporates fine-tuning a pre-trained diffusion model to produce view-specific coordinate maps of canonically oriented 3D objects. Then these aligned geometric priors are integrated into different text-to-image pipelines to generate geometry and appearance.

**Strengths:**

1. The problem addressed in this paper is a realistic problem that current diffusion models struggle to handle effectively.

2. The incorporation of 3D geometric priors into the generation process is a compelling and intriguing approach.

**Weaknesses:**

1. The novelty of this paper is kind of incremental. Most components (such as Stable Diffusion, loss functions) of the proposed method come from previous works, lacking technical novelty.

2. The quantitative results are not satisfying. The evaluation scores reported in Table 1 only show the performance of inconsistency. Did the author compare with other methods by use of other metrics (e.g., R-Precision in DreamFusion)?

3. A simple baseline is missing. Have the authors considered directly replacing the original SDS loss with the aligned geometric SDS loss and calculated the inconsistencies in Table 1?

4. The ablation discussion is absent.

**Questions:**

Please see Weaknesses.

Other questions:

1. How long does the fine-tuning for a stable diffusion model take?
2. How does the efficiency (inference time) of this model compare to other methods?

---

> ### Author Response · Authors · 2023-11-22
>
> **Q: Novelty**
>
> A: Please refer to the common response for our reply.
>
> **Q: A simple baseline (AGP-only) is missing.**
>
> A: Thanks for pointing this out. When generating the 3D results shown in the submission, AGP has actually dominated the geometry modeling in both NeRF- and DMTet-based pipelines, as we set a large weight  (5x \lambda_ori) for \lambda_align at the beginning of the optimization. As demonstrated in Figure 9 of the revised version, the overall shape generated by AGP-only is almost the same as the AGP + DeepFolyed IF SDS, except for minor details. As a result, the qualitative results are the same as the Ours NeRF-based IF. Please refer to Figure 9 in the revision paper for further visual results at each stage in the generation pipeline.
>
> **Q: The metric of R-Precision.**
>
> A： We thank the reviewer for this insightful suggestion.  For more discussion, please refer to the feedback of Reviewer nptq. During the rebuttal, we obtained the numerical results of appearance-related metrics. We report the R-precision as your suggestion. The results are presented below and in Section A.7 and Table 3 in the revised paper.
>
>
> **Q: The ablation discussion is absent.**
>
> A: Thanks for this suggestion. Please refer to the common response for more details on this point.
>
>
> **Q: How long does the fine-tuning for a stable diffusion model take?**
>
> A: Since we only use coarse geometries, i.e., low-resolution 64x64 CCM maps, to fine-tune the SD, the training process is highly efficient. The whole fine-tuning process takes about one day on 8 A100 GPUs for reasonably good results, and the final model used in the submission was trained for two days.
> We added more details about finetune in the revision A.2.
>
>
> **Q: How does the efficiency (inference time) of this model compare to other methods?**
>
> A: Our AGP model is designed to be as lightweight as possible, with the goal of improving the consistency of generated results. The inference time of the fine-tuned SD model is almost identical to that of the original model. And the time for the whole generation depends on the base text-to-3D model, but our model has introduced almost no additional computational time overhead. Specifically, it takes 0.5 hours for our DMTet-based pipeline and 1 hour for our NeRF-based pipeline, using 4 V100 GPUs.

---

> ### Author Response · Authors · 2023-11-23
>
> Your feedback is greatly appreciated! We've strived to address your concerns in our response. As the deadline approaches, should further queries arise, we are fully prepared to answer. Additionally, we respectfully request your kind consideration regarding a potential adjustment in the rating. Your insights are invaluable to us. Thank you sincerely for your time and support!

---

### Official Review · Reviewer_uDHT · 2023-10-31

**Soundness:** 2 fair
**Presentation:** 3 good
**Contribution:** 2 fair
**Rating:** 5
**Confidence:** 5

**Summary:**

This paper proposes SweetDreamer, a new solution to the multi-face Janus problem encountered in the text-to-3D generation pipeline. The core strategy revolves around integrating a Canonical Coordinates Map (CCM) SDS loss. SweetDreamer fin-tunes the Stable Diffusion model to predict Canonical Coordinates Maps (CCM) on the Objaverse dataset, offering Aligned Geometric Priors (AGP). Furthermore, the authors incorporated camera information into the diffusion model. Both quantitative and qualitative assessments show that SweetDreamer successfully mitigates the issues of geometry and appearance inconsistency.

**Strengths:**

1. The idea of introducing view-conditioned information into 2D foundation models like Stable Diffusion to address the multi-face problem is sound. By fine-tuning the 2D diffusion model to predict the Canonical Coordinates Map, it gains an inherent understanding of the basic viewpoint of an object, thus providing additional aligned geometric priors for the text-to-3D pipeline. Additionally, the inclusion of camera information to speed up convergence is a reasonable choice.
2. Human evaluation results indeed validate that SweetDreamer alleviates the multi-faced problem. Furthermore, a user study accentuates SweetDreamer's superiority over previous methods.
3. The presentation is clear and easy to follow. The author gives a thoughtful analysis of the multi-face problem, including “the geometry inconsistency is the key problem compared with appearance inconsistency”.

**Weaknesses:**

1. There are no ablation studies in this paper. For example, the author claims that the SDS loss from the Canonical Coordinates Map can avoid overfitting the 3D training data. However, there are no experiments to support this idea. Moreover, all the design choices in this paper including how to inject the camera information into the diffusion model are not ablated.
2. In fact, introducing a view-dependent SDS loss to tackle the multi-face issue has been proven in Zero-123 and Magic-123, which use a view-aware SDS loss to generate a 3D object. SweetDreamer uses a Canonical Coordinates Map SDS loss rather than RGB SDS loss, the novelty and technique contribution is limited.
3. SweetDreamer uses Objaverse data to fine-tune the Stable Diffusion model to predict the Canonical Coordinates Map. However, defining a canonical coordinate system uniformly across diverse categories is challenging. The authors' choice to whittle down the Objaverse dataset from 800k to 270k, in effect, results in significant data attrition. When compared with techniques that fine-tune based on multi-view images, this could constrain the method's generalizability.

**Questions:**

NA

---

> ### Author Response · Authors · 2023-11-22
>
> **Q: Ablation studies and the generalization ability of the proposed method.**
>
> A: Please refer to the “More ablation studies” and “Technical novelty” in the common response for the ablation studies and generalization ability,
> respectively.
>
>
> **Q: Novelty.**
>
> A: Please refer to the “Technical novelty” in the common response.
>
>
> **Q: Filtering the data and not using all may constrain the generalization ability.**
>
> A: Our filtering process mainly filters out noisy 3D data, such as low-poly shapes, point clouds, unbounded scenes, etc, using some simple and heuristic rules. In addition, we also filtered out some obviously wrongly oriented samples in certain categories (e.g., characters) based on the orientation of their Axis-Aligned Bounding Box (we missed this filtering in the submission and have revised accordingly). While we agree that the data can potential affect the fine-tuning process, we also believe that inherently the key is the diversity and quality of the data rather than simply the absolute amount of data. Moreover, we only use the coarse geometries in the dataset for aligning the diffusion priors at the coarse geometry level, such “less is more” way of using the data has demonstrated its efficacy in alleviating the multi-view inconsistency issues without compromising significantly the generalizability of powerful diffusion priors in terms of highly varied appearance and geometric details, as evidenced by extensive successful results in the submission.
>
> In addition, recent works have also found that 2D diffusion models are already very powerful, and the results of lightweight fine-tuning using a small number of high-quality 3D data may even be better than training with a larger amount of data (such as Wonder3D and Instant3D). This is consistent with our experimental experience, and we have added more discussions on this in the revision (See Appendix A.5).
>
> Last, we have also added a discussion on the generalizability of our method in Appendix A.4. More results, that showcase the ability to generalize to highly varied objects unseen in the dataset, can also be found on this anonymous website. http://3dsweetdreamer.github.io/nerf-based-gallery_0.html.

---

> ### Author Response · Authors · 2023-11-23
>
> Your feedback is greatly appreciated! We've strived to address your concerns in our response. As the deadline approaches, should further queries arise, we are fully prepared to answer. Additionally, we respectfully request your kind consideration regarding a potential adjustment in the rating. Your insights are invaluable to us. Thank you sincerely for your time and support!

---

### Official Review · Reviewer_nptq · 2023-11-01

**Soundness:** 3 good
**Presentation:** 3 good
**Contribution:** 2 fair
**Rating:** 6
**Confidence:** 3

**Summary:**

This paper proposes aligned geometric priors (AGP) to address the multiview inconsistency problem in diffusion models. In detail, the method finetunes a pre-trained 2D diffusion model to generate viewpoint-specific coarse geometric maps of canonically oriented
3D objects to ensure 3D awareness. Qualitative experiments and human evaluation results show great geometry consistency of the proposed method.

**Strengths:**

1. The proposed aligned geometric priors (AGP) show effectiveness in improving multiview consistency.

2. The qualitative results show good consistency compared with other methods.

3. The paper is well-written and the core contributions are clear.

**Weaknesses:**

1. Using Canonical Coordinates Map (CCM) in diffusion model to preserve 3D consistency is one of the core contributions of this paper. However, this coordinate map representation is not something new, and the advantage of using CCM instead of a depth map, normal map, or point cloud is not clear. Experiments to show the superiority of such representation are desired.

2. The experiments have some weaknesses. Only 80 results from 80 text prompts are generated using each method. The results can be noisy based on the small amount of data.

3. The definition of "3D consistency" in the quantitative experiment is vague ("e.g., multiple heads, hands, or legs"). For example, when the generated object is blurred, or does not have the concept of "head, hand, leg" (e.g., building), or some part of the generated object is twisted but not duplicated or missing, how can we determine the 3D consistency is correct or not?

4. It is better to also report the appearance or texture quality measurements, even though this paper mainly focuses on improving the 3D consistency. It can show whether (and if yes, to what extent) the proposed method will influence the appearance or texture quality of the generated objects.

5. The method is based on the assumption that all objects within the same category adhere to a canonical orientation in the training data. However, Objaverse does not align the zero poses of the objects. It is interesting to see whether only using a smaller subset, where objects have a canonical orientation, can lead to better results or not.

**Questions:**

1. The questions in the Weaknesses section.

2. "Ours (NeRF-based full) using DeeFloyd IF first and then Stable Diffusion". How is it implemented in detail? What is the method (Ours (NeRF-based)) used in user study (NeRF-based IF or NeRF-based full)?

3. How are $\lambda^{ori}$ and $\lambda^{align}$ chosen?

4. The user study interface can be shown in the appendix (How did the interface show users to "consider only the 3D consistency"? How are the users chosen? Do they have the concept of 3D consistency?)

I will consider raising the rating if the authors can respond to the weaknesses and questions well in the rebuttal.

---

> ### Author Response · Authors · 2023-11-22
>
> **Q: Ablation studies between CCM and other representations.**
>
> A: Thanks for the valuable comment, please refer to the common response about  “More ablation studies” for more details.
>
>
> **Q: Limited experiments.**
>
> A: Almost all existing text-to-3D methods, including ours, share a common limitation that the optimization-based generation process could take hours for obtaining a single 3D result. For example, DreamFusion takes 1.5 hours on 4 TPUs, Magic3D 40 minutes on 8 GPUs, Fantasia3D 1 hour on 4 GPUs, and ProlificDreamer approximately 8 GPU hours. Due to this very reason, while DreamFusion managed to present results obtained using 398 prompts, the number of results presented in most of these works is indeed limited, such as 40 prompts in MVDream and a dozen in Fantasia3D and ProlificDreamer.
>
> Yet, in our submission, we have presented around 100 results obtained using various arbitrary prompts in the video and supplementary, and compare to baseline methods on 80 prompts. Therefore, we would like to emphasize that our experimental results are sufficient and should not be considered as a weakness, as these prompts are several times more than those used for evaluating previous methods. During the rebuttal, we have increased the number of text prompts up to 150 (see anonymous links http://3dsweetdreamer.github.io/nerf-based-gallery_0.html and https://3dsweetdreamer.github.io/dmtet-based-gallery_0.html) but are indeed limited by the computing resources for more, we are happy to add more results using more prompts to the revision in the next stage.
>
>
> **Q: Criteria for determining 3D consistency in quantitative evaluations.**
>
> A: We agree that the definition of the 3D consistency can be vague to describe, and currently there is a lack of objective metrics that can be computed automatically to determine if a 3D object is multi-view inconsistent or not. This is why we rely on human users for quantitative evaluation of the 3D consistency of the generated results. Specifically, the human participants involved in Table 1 of the paper are researchers from academic laboratories, with highly relevant research backgrounds such as Computer Vision, Computer Graphics and Visual Computing, while the users involved in obtaining the preference rate charts are from loosely related backgrounds such as CS and EE (a more noisy evaluation process, yet our method consistently outperforms other baselines). Furthermore, we have included the user study interface in the appendix of the revised paper for clarification.
>
>
> **Q: Report the appearance quality.**
>
> A: We thank the reviewer for this insightful suggestion. As also mentioned in the common response, our method can preserve the high realism of the 2D results of the powerful text-to-image foundation model, as we only align the geometric priors at the coarse geometry level. This is particularly attractive, as the 3D dataset (e.g., Objaverse) in its current form falls short of covering samples as diverse as the 2D text-image dataset, particularly in terms of the appearance. During the rebuttal, we have obtained the numerical results of appearance-related metrics. As suggested by Reviewer UgSj, we use the R-precision, which is also used in related work. The results are presented below, and in Section A.7 and Table 3 in the revised paper. It is obvious that our method achieves the best performance on this metric over competing methods.
> | Method           | CLIP B/32 R@1 | CLIP B/32 R@5 | CLIP B/16 R@1 | CLIP B/16 R@5 |
> |------------------|---------------|---------------|---------------|---------------|
> | Magic3D          | 60.1          | 71.5          | 62.7          | 77.7          |
> | TextMesh         | 51.7          | 65.1          | 55.1          | 78.4          |
> | SJC              | 40.2          | 51.2          | 52.5          | 62.5          |
> | DreamFusion      | 59.7          | 70.2          | 61.6          | 74.3          |
> | Ours (NeRF-based)| **77.5**      | **84.9**      | **88.7**      | **92.3**      |

---

> ### Author Response · Authors · 2023-11-22
>
> **Q: Whether using a small amount of high-quality data for fine-tuning will improve the performance.**
>
> A:  In the submission, our model is fine-tuned using 3D objects obtained by filtering the Objaverse via some simple and heuristic rules. In addition, we also filtered out some samples in certain categories (e.g., humans) simply based on the orientation of their Axis-Aligned Bounding Box (we missed this filter in the submission and have revised accordingly). After these filterings, most of the data have been canonically aligned (around 80%, estimated based on statistical random samplings), and the rest wrongly oriented objects have only a minor impact on the performance of our model, as manifested by extensive successful results in the submission.
>
> In addition, as suggested, we have also trained our model on a carefully and manually selected dataset, which consists of 20,000 objects. The resulting model is still able to generate highly varied 3D results, with detailed appearances (See A.5 and Figure 10 of the revised paper). While we have not observed significant improvement on this setting, we believe that using more well-structured data could potentially benefit the 3D generation and will perform a more thorough study in the next stage.
>
> **Q: More implementation details.**
>
> A: Ours (NeRF-based full) uses the DeepFloyd IF model first and then using SD. While DeepFloyd IF can be a more powerful model for text-to-image, it can only perform at the 64 x 64 resolution on our side due to its use of much more computational resources. For a fair comparison, Ours (NeRF-based) in the user study is conducted using only DeepFloyd IF, which is the same as other competing methods.
>
>
> **Q: Hyperparameters.**
>
> A: All results presented in the paper were generated using the same hyperparameters, without the need for adjusting individually for each generation. More specifically, at the beginning of the generation we guide the generation to focus more on the geometry, so the $\lambda^{align}$ is set to a larger value (5x $\lambda^{ori}$) and then decreased to 0.5x for higher-quality appearance.
>
>
> **Q: More details about the interface of the user study and the criteria for user selection.**
>
> A: We added more details and the figure of the interface for user study in the appendix. Please kindly see the feedback in the revision.

---

> ### Comment · Reviewer_nptq · 2023-11-22
>
> Thank the authors for the detailed reply.
>
> The ablation studies between CCM and other representations show the advantage of CCM. The new results measuring CLIP R-Precision show the advantage of the proposed method on image quality. Moreover, the added details would be helpful for readers to have a better understanding.
>
> One minor question: in the user study interface, have the generated objects been shuffled in each row? It seems that in the figure of the NeRF-based method, the objects in the 4th row always have the largest size.

---

> > ### Author Response · Authors · 2023-11-23
> >
> > Thanks for your time and kind comment :)  Yes, the generated objects have been shuffled during the user study. In Figure 12, which shows the backend of the user study interface, the objects are in order for easier design. When we distribute the questionnaire, we will shuffle the options.

---

> ### Author Response · Authors · 2023-11-23
>
> Hi, we hope our responses have effectively addressed all your concerns. Please be informed that the author-reviewer discussion period ends today. Should there be any unresolved issues, we would appreciate your prompt feedback. If your concerns have been addressed, please kindly consider updating your score. Thank you!

---

### Official Review · Reviewer_Q6rV · 2023-11-02

**Soundness:** 4 excellent
**Presentation:** 4 excellent
**Contribution:** 4 excellent
**Rating:** 8
**Confidence:** 4

**Summary:**

Text-to-3D is now a very very hot topic, starting from Dreamfushion. Nowadays, there are already many works which demonstrate very good quality of results. However, existing methods undergo a severe issue: the SD lacks view information of the 3D objects, making "multi-face" issues are usually caused. This work presented Canonical Coordinates Map as a geometric representation and train a geometric-aligned prior resorting to the recent public 3D dataset-Objaverse. The results are of very high quality.

**Strengths:**

- First, the results are of very high quality.
- The paper writing is good and the motivation is strong.
- The proposed method is novel.

**Weaknesses:**

- My major concern is about the necessity of the proposed CCM. It seems CCM is one of the geometric prior representation. I am curious what will happen if we replace CCM by just using normal map or depth map as a geometric representation? More specifically, an extra experiment is needed: using objaverse to train a normal/depth map (instead of CCM) generative model and also include camera condition.
- My another question is: if objaverse is used to finetune SD for CCM generation, will the generalization ability be decreased? As objaverse is lacking diversity especially on complicated scenarios. More discussions are encouraged.
- In Fig 1, it seems the normal map owns many bumps, what are the details about the normal rendering?
- It is said in the implementation details "a significant portion of the 3d objects in canonical orientation and a few misoriented". The authors mentioned it will not affect the results. I think more detailed explanations are needed, for example, what does it mean about mis-oriented? how many objects are mis-oriented？ if the mis-oriented cases are corrected manually,  will the performance be improved further?

**Questions:**

see above

---

> ### Author Response · Authors · 2023-11-22
>
> **Q: Ablation studies among CCM, depth map, and normal map.**
>
> A: We appreciate your feedback. Please refer to the common response for more details.
>
>
> **Q: The generalization ability of the model after fine-tuning.**
>
> A: Since we only finetune and constrain the geometric prior at a coarse resolution, the powerful generalization ability of the image diffusion models is largely preserved. The appearance modeling totally relies on the original diffusion model trained on billions of high-quality images. During the geometric modeling process, we were surprised to find that the finetune model still has strong generalization ability and can generate prompts such as Einstein wearing a gray suit and riding a bicycle. This is difficult to achieve with other existing methods. We have added more discussion on the generalizability in Appendix A.4.
>
>
> **Q: More details about normal rendering.**
>
> A: The bumps come from the tangent space normal. For appearance modeling, we follow Fantasia3D and optimize the tangent space normal, which is a crucial component
>  of Physically Based Rendering (PBR) materials. The normal map showcased in our paper emerges from offsetting the original geometric normal through the application of
>  tangent space normal. This procedure generates multiple bumps in visual, thereby enriching both the hierarchical structure and realism of the appearance. It allows the
>  representation of intricate surface details without increasing the geometric complexity of the underlying model. This efficiency is particularly useful for representing
>  high-resolution details on low-resolution geometry.
>
> **Q: More details about the orientation.**
>
> A: Let us illustrate the concept of 'mis-oriented' with an example. In the objaverse dataset, imagine 3D asset labeled as a 'cute panda.' When this asset is rendered from an azimuth=0 perspective, deviating from the front view of the panda, we classify this as the object being mis-oriented. Conversely, if the front view aligns with azimuth=0, we deem this as the correct or consistent orientation. Following the application of a specific set of rules to filter around 270k objects from the initial dataset, we conducted random sampling and manually assessed the percentage of objects exhibiting the accurate orientation. Our findings revealed that roughly 80% of the data exhibited consistent orientations. We reveal that the orientation of objects generated by the proposed model trained with such dataset is almost always correct and has a strong generalization ability. In order to further investigate the impact of data with a higher proportion of correct orientation on the generated results, We then manually screened the top 20k data, resulting in a dataset of 18,488 instances with a uniform orientation and high correspondence with their descriptions. From the experimental results obtained by training on such a smaller dataset, we are surprised to find that even with a smaller dataset of only 20k instances for fine-tuning, our text-to-3D pipeline still exhibits strong generalization and is capable of generating diverse 3D objects. Please refer to Appendix A.5 in the revised version of our paper.

---

> ### Author Response · Authors · 2023-11-23
>
> Your feedback is greatly appreciated! We've strived to address your concerns in our response. As the deadline approaches, should further queries arise, we are fully prepared to answer. Additionally, we respectfully request your kind consideration regarding a potential adjustment in the rating. Your insights are invaluable to us. Thank you sincerely for your time and support!

---

### Author Response · Authors · 2023-11-22
**Response to common questions (1/2)**

We thank all the reviewers for valuable comments and feedback. Here we address the common questions raised by most of the reviewers, mainly including the absence of ablation studies regarding various aspects, and the novelty of our work. During the rebuttal, we have conducted additional experiments and revised the paper accordingly. We briefly summarize these experiments and changes in the following.

# More ablation studies (R#Q6rV, R#nptq, R#uDHT, and R#UgSj)

Regarding the geometric representation, we agree that, besides our CCM, both depth and normal representations can serve a similar purpose of conveying the geometric information during fine-tuning the process. In the rebuttal, we have also derived variants of our method using these geometric representations and obtained the results, which show that, while these two geometric representations can also be incorporated into our pipeline for aligning the geometric priors in 2D diffusion, they indeed fail in producing 3D results as high-quality as CCM.

On one hand, a depth map is a single-channel map storing the per-pixel distance between the surface to the camera. Compared to CCM, the depth map alone does not contain essential information about the relative orientation of the 3D object and the camera, so the network can solely rely on the camera information for learning 3D-aware priors, consequently leading to degraded robustness of the learned 3D priors. In stark contrast, the CCM alone has actually encoded the information of the orientation to some extent. This is manifested by the distinct color coding results of CCMs from different viewpoints. As a result, the variant using depth maps is more prone to multi-view inconsistent issues, compared to our method using CCM.
On the other hand, a normal map stores the local directions of fragments on the 3D surface, we use normal maps obtained from the canonical coordinate space for fair comparisons. Our experimental results show that, while the normal map can also be seamlessly integrated into our pipeline, the text-to-3D optimization process has difficulty in generating a 3D world, of which the derivative follows faithfully the generated normal maps, and may fail to generate meaningful 3D results. Those successfully generated 3D results tend to be slightly smoother regarding the geometric and appearance details of the surface.
In addition, although the point cloud is also an important geometric representation, it is non-trivial to integrate it into the powerful 2D diffusion models as  it is not compatible with the U-Net network architecture. So we could not manage to obtain the results using point clouds during the rebuttal.

Last, we have also conducted an ablation study where the camera information injection is removed from our method. The ablation results demonstrate that removing those extrinsic parameters leads to a significant decrease of the consistency in the generated results, due to the lack of critical camera information.

We have included all these results and discussions into the revised version of our paper (See Section 4 ablation part, A.6, etc).
We also prepared an anonymous link: http://3dsweetdreamer.github.io/ablation.html for better visualization in the 360° video.

---

> ### Author Response · Authors · 2023-11-22
> **Response to common questions (2/2)**
>
> # Technical novelty (R#uDHT and R#UgSj)
>
> We respectfully disagree and believe that our method, particularly with the use of CCM,  is indeed novel in the field of text-to-3D. Novelty does not necessarily mean high complexity or difficulty of the method. Like many research works that build upon the success of previous literature, we also stand on the shoulders of giants and show that the integration of different well-established components such as the SDS loss, Stable Diffusion, and most importantly the novel model fine-tuned with CCMs, can contribute to a text-to-3D pipeline that can effectively address the notorious multi-view inconsistency problem, and equally importantly, retains to the maximum extent the generalizability of the foundation text-to-image model in the pipeline in terms of the highly varied appearance and geometric details.
>
> Notably, we would like to highlight that this preservation of generalizability is particularly attractive, as it can lead to more diverse and highly realistic 3D generation results. And we have achieved this by only aligning the geometric priors in 2D diffusion using the coarse geometric information (CCM) of well-defined geometries, without compromising the original diffusion priors in the text-to-3D pipeline regarding detailed appearances and geometries. While we have validated this through the comparison results against MVDream in the submission (Fig. 7 in the supp.), we have also prepared more qualitative results in the following anonymous link, to further showcase such ability of our method to generalize smoothly to 3D worlds with highly realistic and varied appearance and geometric details, that are unseen in the 3D dataset: http://3dsweetdreamer.github.io/nerf-based-gallery_0.html. We have also added these experimental results in the revised paper (See Section A.4, Figure 7, 8, etc.).

---

### Author Response · Authors · 2023-11-22
**General Response**

We are grateful for the reviewers' constructive comments and insightful feedback, which are instrumental in enhancing the presentation of our work.
We are delighted that all the reviewers recognized the primary contribution of our work and agree that the ‘motivation is strong’ (Reviewer 16rV), with ‘thoughtful analysis’ (Reviewer uDHT), the ‘proposed method is novel’ (Reviewer 16rV), ‘compelling and intriguing’ (Reviewer UgSj), and the results are of ‘high quality’ (Reviewer 16rV) and ‘good consistency’ (Reviewer nptq).

As mentioned by Reviewer UgSj, our work targets a realistic problem that frequently arises in current task-to-3D tasks. We have proposed a generic AGP module that can be seamlessly integrated into various existing pipelines to generate 3D objects of high consistency. The generated results exhibit high consistency and photorealistic appearance, demonstrating state-of-the-art performance in text-to-3D.

We have made necessary revisions to our manuscript to address and elucidate the concerns raised. To summarize, the paper has been improved in the following  aspects:

- Added comprehensive ablation studies on different geometric guidance and the camera embedding module, which have provided convincing evidence that our method exhibits superior performance. (Section 4, A.6, Figure 5, 11 and the anonymous link http://3dsweetdreamer.github.io/ablation.html)
- Discussed and added more results about the generalizability of our proposed method. Our method can produce a wide variety of 3D results with intriguing appearance, greatly preserved the generalization ability of the image generation model. (A.4, Figure 8, 9 and the anonymous link http://3dsweetdreamer.github.io/nerf-based-gallery_0.html and https://3dsweetdreamer.github.io/dmtet-based-gallery_0.html)
- Conducted quantitative evaluations using appearance-related metrics. (A.7 and Table 2)
- Exmplored more on the effect of different settings of dataset. (A.5 and Figure 10)
- Provided the more implemention details in finetuning and optimization.  (A.2)
- Provided more details about the user study, including the interface. (Section 4, Figure 12)

We believe that these enhancements enhance the excellence and comprehensibility of our paper, and we are eager to receive additional feedback from reviewers.

---

### Meta-Review · Area_Chair_TxV9 · 2023-12-13

**Metareview:**

The paper has received diverged reviews with two acceptance and two rejection recommendations with scores 5/6/5/8.

The main concern is the insufficient ablation study compared to other geometric priors (surface normal, depth) and the novelty of adding camera viewpoint aware priors. In the rebuttal, authors have added comprehensive experiments comparing the proposed method to multiple baselines. Reviewer nptq agreed that the additional experiments in the rebuttal improves the manuscript and addresses the concern. The AC concurred with the assessments.

Despite that existing and concurrent works have demonstrated using geometric or multi-view aware priors improve text-to-3d generation, this paper proposes using CCM learned with synthetic data and shows it superior to existing methods. The AC agreed that learning geometric priors from synthetic dataset, instead of RGB multi-view priors, is a novel approach and complementary to the existing works.

Overall, AC found that the rebuttal has addressed most concerns raised in the reviews and concurred that this work has novel contribution to the 3d generation field. Consequently, AC recommends accepting the paper and urges authors to add additional studies into the final revision.

**Justification For Why Not Higher Score:**

Despite that authors propose a novel geometric priors complementary to existing works, it does not propose fundamental innovations over existing methods. Consequently, AC recommends accepting the paper with poster.

**Justification For Why Not Lower Score:**

Most concerns in ablation study and novelty have been addressed by the additional ablation study in the rebuttal. The AC does not find enough evidence to overturn the acceptance recommendations from reviewers.

---

### Decision · Program_Chairs · 2024-01-16

Accept (poster)